# Total Antioxidant Capacity of Saliva and Its Correlation with pH Levels among Dental Students under Different Stressful Conditions

**DOI:** 10.3390/diagnostics13243648

**Published:** 2023-12-12

**Authors:** Christoph Schwarz, Octavia Balean, Ramona Dumitrescu, Paula Diana Ciordas, Catalin Marian, Marius Georgescu, Vanessa Bolchis, Ruxandra Sava-Rosianu, Aurora Doris Fratila, Iulia Alexa, Daniela Jumanca, Atena Galuscan

**Affiliations:** 1Translational and Experimental Clinical Research Centre in Oral Health, Department of Preventive, Community Dentistry and Oral Health, University of Medicine and Pharmacy “Victor Babes”, 300040 Timisoara, Romania; christoph.schwarz@umft.ro (C.S.); balean.octavia@umft.ro (O.B.); dumitrescu.ramona@umft.ro (R.D.); bolchisvanessa@gmail.com (V.B.); jumanca.daniela@umft.ro (D.J.); galuscan.atena@umft.ro (A.G.); 2Clinic of Preventive, Community Dentistry and Oral Health, Department I, University of Medicine and Pharmacy “Victor Babes”, Eftimie Murgu Sq. no 2, 300041 Timisoara, Romania; 3Department of Biochemistry and Pharmacology, Victor Babeş University of Medicine and Pharmacy, Pta Eftimie Murgu Nr. 2, 300041 Timisoara, Romania; paula.muntean@umft.ro (P.D.C.); cmarian@umft.ro (C.M.); 4Functional Sciences Department, Physiology Discipline, Victor Babes University of Medicine and Pharmacy of Timisoara, 2 Eftimie Murgu Sq., 300041 Timisoara, Romania; georgescu.marius@umft.ro; 5Faculty of Dental Medicine, Ludwig-Maximilian-University Munich, Goethestraße 70, 80336 München, Germany; a.fratila@campus.lmu.de; 6Department of Dentistry, Faculty of Dental Medicine, “Vasile Goldis” Western University of Arad, 310045 Arad, Romania; alexa_iulia@ymail.com

**Keywords:** saliva, total antioxidant capacity, pH, STAI test, stressful condition

## Abstract

(1) Background: This cross-sectional study conducted at the Faculty of Dental Medicine, Timisoara, Romania, between December 2022 and February 2023 aims to assess salivary total antioxidant capacity and pH levels in dental students experiencing non-stressful and stressful situations and explore potential correlations between these factors. (2) Methods: Saliva samples were collected during two different periods: before an Oral Health course and before the Oral Health exam, under stressful conditions. Ethical principles were followed, and informed consent was obtained. Data on age, gender, health status, drug use, smoking habits, and anxiety levels were recorded. Saliva was collected using the draining method and pH was measured using indicator paper strips. Total antioxidant capacity (TAC) was determined using a commercial assay kit. Statistical analysis involved descriptive statistics, Student’s *t*-test to compare pH and TAC between study groups, and Pearson’s correlation coefficient to analyze the correlation between salivary pH and TAC within each group, with *p* < 0.05 indicating significance. (3) Results: This study involved 80 participants, comprising 26 males and 54 females, all enrolled in the 5th year of the Oral Health course, with ages ranging from 20 to 53 and a mean age of 23.62 (±4.19) years. Pearson’s correlation results show a statistically significant negative relationship between the STAI test and TAC during the stress-free period (−0.02 **, *N* = 80, *p* < 0.01). (4) Conclusions: There are variations in saliva’s antioxidant capacity in response to different stress conditions. Dental students experienced a higher level of stress before academic assessments compared to the non-stress period during the course.

## 1. Introduction

Interest in rapid and less invasive diagnostic tests has grown exponentially in the recent decade, which has led to extensive research on saliva as a biological fluid for clinical diagnosis [1,2]. Saliva is currently recognized as a potential reservoir for various biological markers, encompassing alterations in biochemicals, DNA, RNA, proteins, and the microbiota structure. Collecting saliva is a relatively safe method that minimizes the risk of virus transmission. Therefore, saliva emerges as a novel, non-invasive, and straightforward approach to assist in disease diagnosis, with the anticipation that it will eventually serve as a viable alternative to traditional serum or urine tests in diagnostic processes [3,4]. Saliva-based tests have been successfully used in human immunodeficiency virus (HIV) infection diagnosis [5], monitoring renal disease [6], prevention of cardiometabolic risk [7], detection and quantification of viral nucleic acids [8], dental studies [9,10] and drug abuse monitoring [11]. There are also some studies proposing the use of saliva in monitoring physically active individuals, incremental effort test [1,12,13], and psychological stress [14].

To acknowledge the significance of saliva as a diagnostic fluid, the New York Academy of Sciences took the initiative to sponsor a significant conference on this subject in 1992 [15]. During the conference, participants emphasized the need for the advancement of highly sensitive and specific assays to effectively measure and comprehend the variations in saliva related to drug therapy, drug abuse, endocrine function, systemic and oral diseases, genetic defects, nutritional status, and age-related changes. This conference played a vital role in raising awareness regarding the potential of saliva-based diagnostics. As a result, continuous research efforts have paved the way for the development of more sensitive salivary assays, thus deepening our understanding of the interconnection between oral health and overall wellbeing. Oxidative stress refers to the disruption between the generation of reactive oxygen species (ROS) and the body’s ability to counteract them with antioxidants. This imbalance within the human body is a notable risk factor that contributes significantly to the development of noncommunicable diseases [16]. Total antioxidant capacity (TAC), defined as the moles of oxidants neutralized by one liter of solution, is a biomarker measuring the antioxidant potential of the body’s fluids [17].

In recent years, there has been a shift in focus among clinicians and researchers towards recognizing the importance of the antioxidant capacity of saliva as a crucial line of defense against chronic degenerative diseases. Saliva, with its unique composition and properties, has gained recognition for its potential role in protecting against the detrimental effects of oxidative stress. Understanding and harnessing the antioxidant capacity of saliva could offer promising avenues for preventive and therapeutic strategies in combating chronic degenerative diseases [18]. From an ethical standpoint, saliva can be regarded as the ideal research material for scientific investigations involving humans. Saliva serves as a highly convenient diagnostic biological fluid due to its unique properties. Notably, saliva does not coagulate, making it easy to handle in laboratory settings. Furthermore, it remains stable for diagnostic purposes for up to 24 h at room temperature and can be stored for a week at 4 °C, enabling flexibility in sample collection and analysis [19]. Human saliva can be collected multiple times a day, making it easier to conduct repeated analysis for therapy monitoring. Compared to blood, saliva is a superior diagnostic screening material because it avoids the anxiety associated with blood collection. The more tolerable nature of saliva collection may lead to a reduced reluctance in visiting diagnostic biochemical laboratories, enabling earlier diagnostics and potential cost savings in the healthcare budget [20].

Every biological fluid, including saliva, contains various antioxidant mechanisms that are always ready to deal with different etiologies of stressors. The phrase “mirror of the body’s health”, used to describe saliva, illustrates how valuable saliva is as a source of parameters for the body’s biological response to psychological stress [21]. Elevated levels of acidity in saliva are recorded when a person is in a state of anxiety. During stress, fear, and anxiety, saliva secretion decreases, and the concentration of hydrogen ions (pH) promotes acidity [22].

Over 50% of medical and dental students reported experiencing stress. Stress varies among individuals and is impacted by interpersonal, intrapersonal, intellectual, and environmental factors. This can be detrimental as it places additional strain on their physical, mental, and emotional wellbeing [23]. Academic exams represent a tangible example of major stressors in students’ lives, as they are time-constrained and typically perceived as aversive or threatening. The examination experience is laden with a stressful and daunting atmosphere for most of the students, which is why this type of stressor is often studied as a model for psychological and physiological reactions observed in stressful situations. Test-related anxiety encompasses the specific emotional and physiological responses evoked by the testing stimulus and includes cognitive components (i.e., worry) and emotional and physiological arousal components (i.e., emotional excitement) [24].

The Spielberger State–Trait Anxiety Inventory (STAI) was created by Spielberger et al. in 1970 to evaluate anxiety levels based on both temporary states and enduring traits. The state measurement assesses an individual’s present moment anxiety levels, asking them to rate the intensity of their feelings on a four-point scale ranging from “not at all” to “very much so”. On the other hand, the trait anxiety measure explores how individuals generally experience anxiety by rating themselves on a four-point scale from “almost never” to “almost always” [25].

This study presents a comparative evaluation and correlation analysis of the salivary total antioxidant capacity and pH among dental students experiencing stressful conditions. Dental students are subjected to various stressors during their academic journey, and these stress conditions can potentially impact their oral health.

Thus, the objectives were to determine the impact of psychological stress on selected salivary parameters (salivary pH) and to assess the correlation between stress and oxidative parameters. For patients seeking dental care, it is of utmost importance to consider indicators such as stress resistance, anxiety levels, diagnostic criteria by which these factors are evaluated, treatment choices, materials applied, and methods used to correct salivary pH changes caused by anxiety as they enable the enhancement of oral health quality.

To date, there has been no study conducted in Romania examining the salivary antioxidant capacity of students under various stress conditions and its correlation with salivary pH. By exploring this association, valuable insights can be gained regarding the impact of stress on oral health and the potential protective role of salivary antioxidants.

Understanding the relationship between salivary parameters and stress conditions in dental students can provide valuable insights into the impact of stress on oral health and potentially contribute to the development of preventive measures and interventions.

## 2. Materials and Methods

This cross-sectional study was carried out between December 2022 and February 2023 in a single center (Translational and Experimental Clinical Research Centre in Oral Health, Department of Preventive, Community Dentistry and Oral Health, University of Medicine and Pharmacy “Victor Babes”, Timisoara, Romania) on a group of students attending the Faculty of Dental Medicine, during two different periods. The first saliva collection took place prior to an Oral Health course in which the students participated during the semester. The second collection occurred before the start of the Oral Health exam, under stressful conditions, and during the exam session. This study adhered to the ethical principles outlined in the World Medical Association Declaration of Helsinki (1964). Prior to conducting this study, permission was obtained from the Ethical Committee of the University of Medicine and Pharmacy “Victor Babes”, Timisoara, Romania (no. 34/2018). Participation in the study was strictly voluntary, and all individuals included in this study provided their informed consent by reading and signing a consent form.

The selection process involved the participation of 110 students enrolled in the “Oral Health” course, a mandatory component of the curriculum in the fifth year of study at the Faculty of Dental Medicine, UMF, “Victor Babes”, Timisoara, Romania. Attendance for this course is obligatory, and students are required to attend in order to be eligible to take the exam during the examination session. Out of the 110 enrolled students, 80 of them met the participation criteria.

The inclusion criteria required that on the day of saliva sample collection, these students abstained from food consumption for 2 h, refrained from various oral hygiene practices, such as brushing their teeth, and smokers refrained from smoking for one hour prior to the collection. Additionally, they completed a questionnaire gathering data on personal information, health status, smoking habits, and the STAI questionnaire.

Students who did not comply with these mentioned rules or refused to complete the questionnaire were excluded from this study.

During the collection of samples from the study participants, relevant data such as age, gender, general state of health, drug administration, smoking duration (if applicable), and the number of cigarettes smoked per day were recorded.

### 2.1. Assessment of Stress Level

To gauge the extent of stress experienced by the students, they completed the State–Trait Anxiety Inventory. This questionnaire involves brief expressions and serves as a self-assessment tool. Initially designed to examine anxiety in typical adults, subsequent trials demonstrated its applicability to high school students and individuals dealing with psychiatric and physical disorders. The State–Trait Anxiety Inventory comprises two components: the State Anxiety Inventory and the Trait Anxiety Inventory [26]. The State Anxiety Inventory assesses transient emotional reactions known as state anxiety, which are exhibited by individuals in response to non-constant situations. The intensity of these reactions varies depending on the perceived threat level in a given situation. When an individual perceives a situation as stressful or threatening, the level of “state anxiety” is high; conversely, when the perceived danger is not considered threatening, the level of state anxiety is low.

This inventory requires individuals to articulate their current feelings and emotions in specific circumstances, considering their perceptions of the ongoing situation. It gauges the individual’s current anxiety level, reflecting the intensity of their emotions in the present moment. The immediate stress, anxiety, and excitement responses triggered by conditions can fluctuate over time, and individuals respond to the inventory items based on the severity of their emotions at that particular moment.

The Trait Anxiety Inventory involves individuals expressing their general feelings. It assesses anxiety based on how individuals feel “often” and “constantly”. This inventory gauges an individual’s inclination to perceive and interpret situations, typically considered neutral by objective criteria, as threatening and stressful. Respondents rate each item on a 4-point scale (not at all, somewhat, moderately, very much), reflecting the frequency of their generally felt emotions.

The score range for both the STAI-T subscale and the STAI-S subscale falls between 20 (minimum) and 80 (maximum). STAI scores are commonly categorized as follows: “low or no anxiety” (20–37), “moderate anxiety” (38–44), and “high anxiety” (45–80) [26,27].

### 2.2. Collection of Saliva

During this study, the collection of unstimulated whole saliva from subjects was performed using the draining method, which involved collecting saliva until reaching a volume of 2 to 3 mL in sterile tubes (nerbe plus GmbH & Co., Winsen, Germany). Among the various methods available, the draining method was chosen due to its high acceptability for unstimulated saliva collection. All the necessary protocols for saliva collection were meticulously followed and implemented to ensure accurate and reliable results [28].

In order to minimalize the potential impact of circadian rhythms on salivary biochemical determinations, saliva samples were collected specifically between the hours of 8 and 10 AM. This timeframe was chosen to standardize the collection process and minimize any variations that may arise due to natural fluctuations in saliva composition throughout the day. By adhering to this collection window, this study aimed to ensure more consistent and reliable results in the biochemical analysis of saliva [29].

The students abstained from consuming food or beverages (except for pure water) and refrained from performing any oral hygiene procedures, such as teeth brushing, for a minimum of two hours prior to the collection of saliva samples. Smokers were additionally restricted from smoking within one hour prior to saliva collection. Following the guidelines recommended by clinical protocols for saliva collection, the students refrained from taking any medications for at least 8 h prior to the collection of saliva samples. This precaution was taken due to the potential influence of various drugs on salivary secretion [30]. Saliva was collected in the classroom and in the examination room, with the students in a seated and relaxed position, their head slightly tilted downward, minimizing facial and lip movements, after 5 min of adaptation to the environment.

### 2.3. pH Screening of the Saliva Samples

After the collection of saliva samples, the pH determination was promptly carried out using pH indicator paper strips (Qualigens, Glaxo India Ltd., Mumbai, India). The pH values were determined by comparing the color change of the saliva samples to a gold standard chart provided by the manufacturer. The convenience and simplicity of using test strips were advantageous in our case, given the nature of our research and the resources available.

### 2.4. Determination of TAC of Saliva

To ensure the integrity of the samples, they were stored in a container at a constant temperature of 4 °C and transported to the laboratory for total antioxidant capacity (TAC) estimation within one hour of collection.

The total antioxidant capacity (TAC) was determined using a commercially available assay kit (ab65329, Abcam, Cambridge, UK). The kit is based on converting Cu^2+^ to Cu^+^, using standardized to Trolox equivalents (a known antioxidant). Saliva samples were centrifuged at 2500× *g* for 10 min at 4 °C in order to remove debris and other impurities, and then the TAC was determined according to the manufacturer’s instruction using a GloMax^®^ Discover Microplate Reader (Promega, Madison, WI, USA).
Sample Total Antioxidant Capacity (TAC) = (Ts/Sv) × D
where Ts represents the TAC amount in the sample calculated from the standard curve (nmol).

Sv = sample volume added in the sample wells (µL).

D = sample dilution factor.

The workflow diagram can be seen in Figure 1.

### 2.5. Statistical Analysis of the Obtained Results

The statistical analysis was performed using SPSS v23 (Statistical Package for Social Science, IBM, Chicago, IL, USA). Descriptive statistics, including the calculation of mean and standard deviation, were conducted for the salivary pH and total antioxidant capacity (TAC) of saliva. To compare the salivary pH and TAC between the two study groups, i.e., group I (dental students without stress conditions) and group II (dental students under stress conditions), an unpaired Student’s *t*-test was employed. The distribution was checked by calculating the value of kurtosis for each scale we used in both conditions (stress present/non-stress). We considered a normal distribution for the scale because, as stated in the book Laboratory Statistics Methods in Chemistry and Health Sciences, Second Edition, 2018 [31], a standard normal distribution has kurtosis of 3 and is recognized as mesokurtic. Additionally, Pearson’s correlation coefficient was utilized to analyze the correlation between the salivary pH and TAC of saliva within each of the two study groups. For all analyses, *p* < 0.05 was used to assess overall differences.

## 3. Results

The group of participants consisted of 80 students (26 males and 54 females) attending the Oral Health course in the 5th year of study at the Faculty of Dental Medicine, University of Medicine and Pharmacy “Victor Babes”, Timisoara.

The age of the study participants ranged between 20 and 53 years and the mean age was M = 23.62 (±4.19).

### 3.1. Descriptive Analysis of Responses Collected through Questionnaires

Out of the 80 students surveyed, 22.5% (*N* = 18 students) reported having acute or chronic health conditions. Respiratory ailments accounted for 16.25% (*N* = 13) of the reported conditions, including three cases of bronchial asthma, one case of chronic allergic rhinitis, and one history of tuberculosis. Digestive system disorders constituted 5% (*N* = 4) of the reported conditions, with one case of ulcer and one case of gastritis. Additionally, three participants (3.75%) disclosed having endocrine disorders, while four participants (5%) were affected by psychological conditions. Among the latter, two students reported panic attacks, while the other two students suffered from depression.

Furthermore, six of the participants were wearing a fixed orthodontic appliance (metallic), and one participant was wearing a fixed aesthetic orthodontic appliance.

In terms of smoking habits, out of the 80 students surveyed, 51.25% (*N* = 41) identified themselves as smokers. Among these, 30 (37.5%) expressed a preference for traditional cigarettes, while 11 (13.75%) reported using electronic cigarettes.

According to the data collected through administered questionnaires, the quantity of smoked cigarettes varies between 2 and 20 cigarettes per day, with a percentage of 14.4% (*N* = 11) of smokers reporting smoking 20 cigarettes per day.

### 3.2. Descriptive Analysis of pH Results

The salivary pH values recorded during the absence of the stress-inducing factor indicate that 34 out of the 80 subjects had a pH of 7.0, 27 of them had a pH equal to 6.5, and 12 subjects had a pH of 6.0. A pH of 7.5 was observed in 6 subjects, while 1 person had a pH of 8.0.

Salivary pH values underwent changes during the stressful period. Therefore, the majority of the participants had a pH of 6.5 (31 subjects), while 25 students had a pH of 7.0, 3 subjects had a pH of 5.5, 16 of them had a pH equal to 6.0, and 5 subjects had a pH of 7.5.

### 3.3. Descriptive Analysis of TAC Results

As can be observed in Figure 2 and Figure 3, the TAC values between the two recordings underwent changes. During the non-stressful period, the obtained values ranged between 0.84 to 7.24 mmol Trolox Equiv./µL, with most of the examined individuals having values between 1.45 and 3.8 mmol Trolox Equiv./µL. During the stressful period, the values fell within the range of 1.32 and 6.52 mmol Trolox Equiv./µL.

### 3.4. Descriptive Analysis of STAI Results

Regarding the results of the STAI questionnaire applied, it can be observed, according to Figure 4 and Figure 5, that there are value differences between the recorded responses (state and trait) in the stress-free period and the period leading up to the exam session. Regarding the overall STAI test results, the minimum reported score in the stress-free state is 46, and the maximum score is 132, with most values clustered around 84. However, in the period leading up to the exam session, the minimum score obtained for the total STAI test was 51, and the maximum score was 148.

By comparing the results obtained between the two pH measurements, it is observed that the mean values have not undergone significant changes. Specifically, during the non-stressful period, the mean value was 6.7, while it was 6.6 during the stressful period, as shown in Table 1.

### 3.5. Correlations between STAI Test, pH, and TAC Values in Different Stress Conditions and Acute/Chronic Illness Conditions

Pearson’s correlation results show a statistically significant negative relationship between the STAI test and TAC during the stress-free period (−0.02 **, *N* = 80, *p* < 0.01). Additionally, a significant correlation of 0.02 (*N* = 80, *p* < 0.05) is observed between an acute/chronic illness condition and STAI during the exam period. There is also a statistically significant positive correlation between the presence of an acute or chronic illness and the pH level 0.03 ** (*N* = 80), *p* < 0.01 during the exam period, as well as with the administration of treatment 0.04 ** (*N* = 80), *p* < 0.01 (Table 2).

### 3.6. Correlations between STAI Test, pH, and TAC Values in Different Stress Conditions and Smoker Conditions

The research results show a significant positive correlation between the quantity of tobacco consumed and the total STAI score during the non-stressful period (0.03 **, *N* = 80), as shown in Table 3 Additionally, there are statistically significant negative correlations between non-stress TAC and total non-stress STAI (−0.02 *, *N* = 80), as well as total stress STAI (−0.02 *, *N* = 80).

## 4. Discussion

Saliva plays numerous roles within the oral cavity, and its complexity makes it nearly impossible to recreate from individual components. Saliva provides both continuous static protective effects and dynamic effects that operate in response to specific challenges over time. This study aimed to assess the alterations in the total antioxidant capacity of saliva and examine its correlation with pH levels among dental students experiencing varying stressful conditions.

Academic exams serve as an illustration of naturalistic stressors, characterized by their time constraints and typical aversive perception. Academic stress can be described as the daily stress experienced by students, which has repercussions on various aspects of their mental and physical health [32]. Prior research has observed that academic stress levels tend to be higher among younger students in comparison to their older counterparts, particularly in relation to academic responsibilities and concerns such as grades, exams, competition for grades among peers, and the fear of failing the academic year [33]. Examinations are widely regarded as one of the most intense sources of stress for students. Research examining stress responses in students is a prevalent topic in the academic literature, and studies focusing on the influence of academic stress on students’ reactions have contributed valuable data [23]. Prior research demonstrates that the extent of the connection between academic self-efficacy, anxiety, and psychological stress is influenced by several factors, including the nature of the academic challenge (e.g., oral exams) and the presence of an audience. While the outcomes of different studies may vary, certain studies have indicated a noteworthy correlation between academic stress and a reduced immune response on the one hand, and specific aspects of the immune system on the other [34]. Numerous investigations into academic stress have been carried out within the realms of medical and dental education, potentially owing to the fact that many researchers in this field are medical professionals who interact with students.

The stress experienced by medical and dental students is primarily linked to the demanding and challenging nature of their educational programs. Medical and dental school undergraduate programs are often among the lengthiest and most rigorous, leading to significant stress symptoms [23]. Consequently, medical and dental students frequently report heightened levels of anxiety, recurrent depression, obsessive compulsive disorders, interpersonal sensitivity, and other psychological issues. They are frequently examined as a model for understanding physiological responses in stressful situations. In our research, we did not specifically investigate outcome differences between male and female participants, while some studies suggest that there is no statistical significance between male and female participants [22].

In a state of health, the pH of saliva typically remains between 6.7 and 7.4. The current findings indicate that saliva pH levels underwent changes between the two recordings, namely during the semester course (pH 6.73) compared to the exam period (6.58), with the average pH value being close to neutral. While we acknowledge that pH identification is more accurate with a pH meter, the pH values were measured using pH test strips in our study, a method employed in other recent research studies as well [18,35,36]. Considering factors such as smoking, overall health, and drug administrations, pH levels at both time points were linked to the perception of the exam situation’s threat and the experienced stress level with the emotional aspect of exam-related stress.

The current research results are in accordance with the findings from a recent study [37] which illustrated a dose–response relationship between the experienced stress level and pH: as participants’ stress levels increased, their pH levels decreased. Furthermore, stress levels decreased, and pH levels increased during the non-exam period. The pH measurements in this study align with findings from previous research. The authors posit that saliva, being readily accessible, can serve as a suitable fluid for assessing the levels of stress and relaxation [38].

Earlier research has indicated a connection between academic exams and changes in neuroendocrine and immune functions [39,40]. In our study investigating changes in pH (acidity) and TAC (total antioxidant capacity), it was imperative to address the inclusion of participants who had reported acute or chronic illnesses in their questionnaires. While a small portion of our participants had disclosed the presence of such health conditions, we intentionally chose not to exclude them from this study. This decision was driven by our objective to comprehensively monitor the pH and TAC levels even in the presence of these medical conditions. By not excluding participants with health conditions, we aimed to gain a more comprehensive understanding of how pH and TAC may be influenced in diverse stressful contexts. This inclusivity allowed us to explore potential associations between health conditions and pH/TAC variations. Some acute or chronic illnesses are known to affect pH and TAC levels, and by including participants with such conditions, we may have uncovered insights that have clinical implications, such as guiding interventions or treatments. In line with the findings of our study, the total antioxidant capacity (TAC) values exhibited an increase (TAC mean value 3.74) during the stress period compared to the non-stress period (TAC mean value 2.41), aligning with similar research studies.

In modern society, the role of stress and its impact on the body has undergone a shift. While stress reactions were once a protective response to help the body adapt to challenging situations, they now have an increasingly detrimental effect that surpasses the limits of coping. These stress reactions, along with the diseases they contribute to, provide further evidence of the ongoing interplay between physical, psychological, and social factors in health and disease. This highlights the importance and necessity of adopting a multidisciplinary approach for evaluating and treating certain conditions. The field of stress and stress reactions has greatly benefited from studies that utilize specific quantitative and qualitative analyses of saliva. Measuring individual antioxidants in biological samples is a time-consuming, labor-intensive, and costly process that involves intricate chemical techniques [41]. Additionally, considering the additive nature of antioxidant effects, the preferred approach is to measure the total antioxidant capacity of samples. Age and gender differences were observed in oxidative stress markers in plasma [42]. On the other hand, it has been observed that salivary TAC was affected by emotional and psychological factors [43], with these results being in accordance with the changes in TAC observed in our study.

Extensive investigations within the realm of immunology are facing ongoing challenges in uncovering whether and to what degree physiological stress stimuli are linked to biological responses. To our current understanding, there is a limited body of literature addressing alterations in the salivary oxidative status in response to purely short-term acute psychological stress. In a prior study, it was observed that the salivary antioxidant levels in children, assessed using a variety of parameters, exhibited an increase prior to a tennis competition, and this increase was linked to psychological stress [44].

The balance between oxidants and antioxidants is influenced by various personal factors, including environment, diet, physical activity, lifestyle, and metabolism. Total antioxidant capacity (TAC) is notably reduced in cases of periodontal disease, diabetes, smoking, and oral cancer. Exposure to cigarette smoke diminishes the TAC of saliva, turning it into a highly pro-oxidant environment [45]. For this reason, our study also aimed to record whether participants were smokers or non-smokers in order to monitor the changes in TAC. By addressing observed shortcomings and standardizing collection and analysis procedures, saliva has the potential to become a reliable and equally valuable additional diagnostic medium. It is expected to have a significantly greater presence in everyday clinical practice in the near future.

One primary strength of this study is the utilization of the STAI questionnaire, which demonstrates good psychometric properties, such as validity and reliability. The STAI questionnaire is an accredited and piloted questionnaire in Romania, recommended and approved by the College of Psychologists in our country. Another advantage lies in its application method and the time required for completion. The STAI provides objective results for quantifying both anxiety states and general anxiety traits in a very short timeframe (which can contribute to obtaining highly reliable and valid results, as the subject does not become bored, as can happen with lengthy tests). Additionally, the items are formulated in a simple and clear manner to be understood by a wide range of subjects. Moreover, it offers good differentiation between anxiety symptoms and other psychopathological experiences, particularly in distinguishing anxiety from depression. This instrument can be used both clinically for diagnosis and in research, encompassing clinical and non-clinical samples. A recent literature review on psychological stress among students and its assessment using salivary biomarkers [23] reveals that there are other studies utilizing the STAI questionnaire [46,47].

A significant advantage of this study lies in the choice of the biological sample under investigation, which is saliva. The collection of saliva is a relatively straightforward and non-invasive technique, allowing the retrieval of a valuable source that can be utilized to measure various biological factors associated with the stress response.

However, the inventory also has certain limitations. There are studies that question specific types of validity, especially due to the lack of clear differentiation between anxious traits within the anxiety trait scale and depressive traits (partly caused by the high correlations obtained between the STAI scales and other depression scales).

Furthermore, it is worth noting that gender could potentially exert a notable impact on the outcomes, particularly considering that approximately two-thirds of the participants were female. Regrettably, due to the relatively small sample size, we lacked the statistical power to separate and contrast the results based on the participants’ gender.

In addition, conducting pH assessments more frequently could yield additional insights into the relationship between pH, TAC, and exam-induced stress. Another notable limitation of this study is the reliance on a single-point pH measurement for saliva. Since pH levels naturally fluctuate throughout the day, particularly in response to various stimuli and subsequent changes in saliva flow rates, it is reasonable to assume some degree of pH variability. Despite our efforts to minimize pH variability by conducting assessments between 8 and 10 am, after a minimum of two hours of fasting and refraining from smoking, it is possible that conducting multiple assessments within a single day or over consecutive days could provide a more effective control over the influence of pH fluctuations.

## 5. Conclusions

In conclusion, based on the STAI test, TAC, and pH recordings, our study suggests the occurrence of changes in the antioxidant capacity of saliva under different stress conditions. Dental students experienced a higher level of stress before academic assessments compared to the non-stress period during their course. As stress remains a concern even post-graduation, these findings may aid in mitigating the impacts of stress during the transition to professional life. Further research on larger student samples is warranted to obtain more specific data on stress-related salivary biomarkers. Such studies would offer a deeper understanding of the practical utility of interpreting these biomarkers.

## Figures and Tables

**Figure 1 diagnostics-13-03648-f001:**
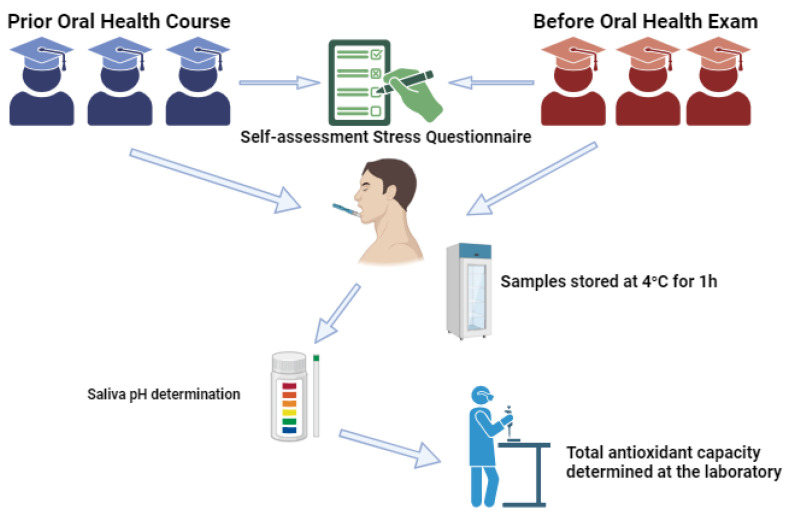
Workflow of data collection and saliva examination.

**Figure 2 diagnostics-13-03648-f002:**
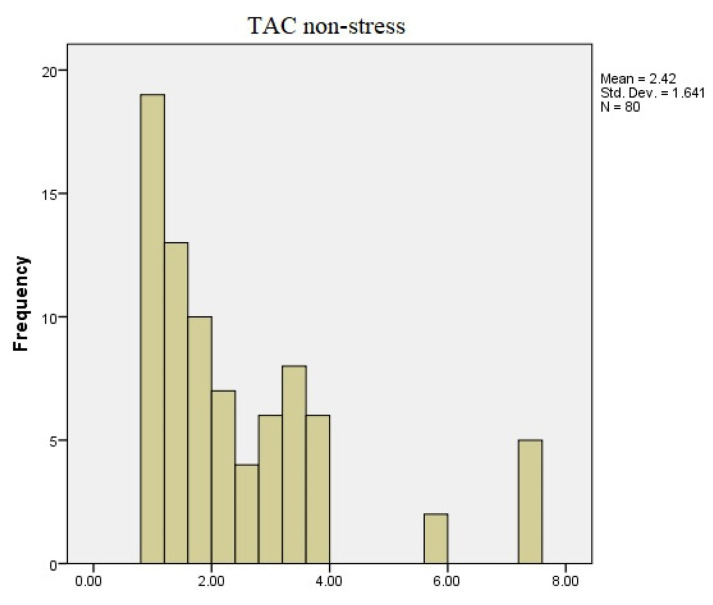
TAC values during non-stressful conditions.

**Figure 3 diagnostics-13-03648-f003:**
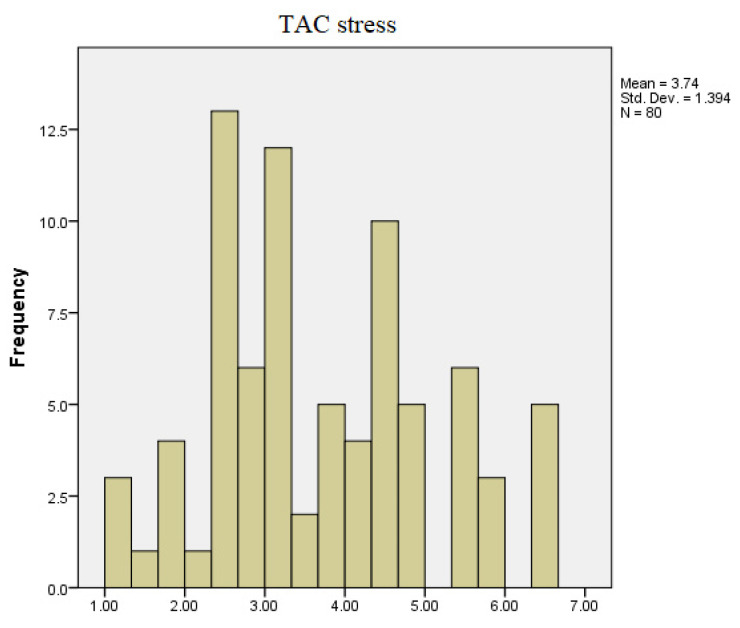
TAC values during stressful conditions.

**Figure 4 diagnostics-13-03648-f004:**
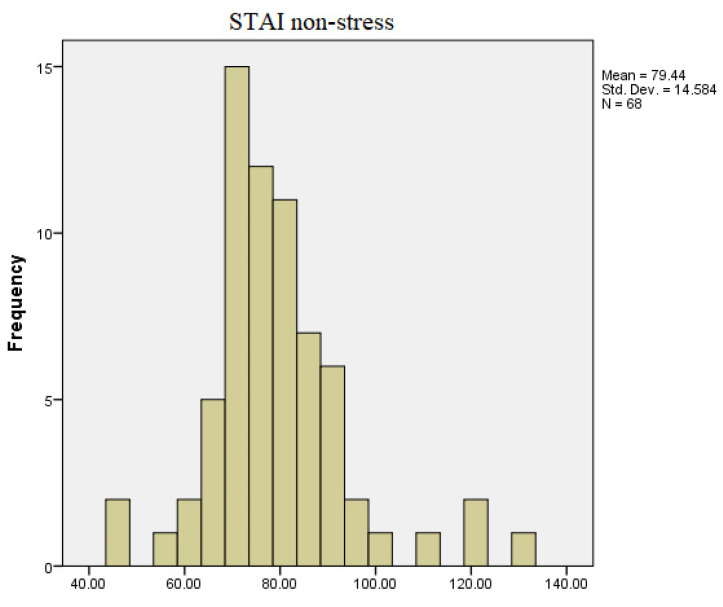
STAI test in non-stressful conditions.

**Figure 5 diagnostics-13-03648-f005:**
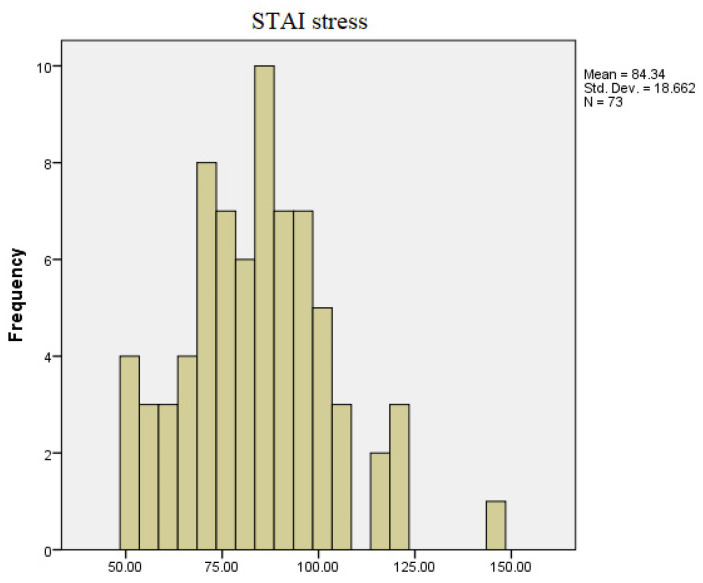
STAI test in stressful conditions.

**Table 1 diagnostics-13-03648-t001:** pH, TAC, and STAI TOTAL values in non-stressful and stressful conditions.

	pH: Non-Stress	pH: Stress	TAC: Non-Stress	TAC: Stress	STAI TOTAL: Non-Stress	STAI TOTAL: Stress
			(mmol Trolox Equiv./µL)	(mmol Trolox Equiv./µL)		
*N* Valid	80	80	80	80	80	80
Mean	6.7	6.6	2.41	3.74	79.44	84.34
Median	7.0	6.5	1.84	3.34	77.00	85.00
Std. Deviation	0.46	0.47	1.64	1.39	14.58	18.66
Kurtosis	1.45	−0.33	2.58	−0.71	2.97	0.81

**Table 2 diagnostics-13-03648-t002:** Correlations between STAI test, pH, and TAC values in different stress conditions and acute/chronic illness conditions.

	Antibiotics	Chronic/Acute Disease	Treatment
pH: non-stress	0.00 (0.07, *N* = 80)	0.01 (0.02, *N* = 80)	−0.00 (0.06, *N* = 80)
TAC: non-stress	−0.01 (0.01, *N* = 80)	−0.01 (0.03, *N* = 80)	−0.01 (0.01, *N* = 80)
STAI: nonstress	0.01 (0.01, *N* = 80)	0.02 (0.00, *N* = 80)	0.01 (0.02, *N* = 80)
pH: stress	0.01 (0.03, *N* = 80)	0.03 ** (0.00, *N* = 80)	0.00 (0.04, *N* = 80)
TAC: stress	0.00 (0.07, *N* = 80)	−0.00 (0.09, *N* = 80)	−0.01 (0.01, *N* = 80)
STAI: stress	0.01 (0.02, *N* = 80)	−0.01 (0.03, *N* = 80)	−0.00 (0.09, *N* = 80)

Pearson’s correlation, sig. (2-tailed), *N*; ** *p* < 0.01.

**Table 3 diagnostics-13-03648-t003:** Correlations between pH, TAC, and STAI (test) in different stress conditions and smoker conditions.

	Time Being Smoker	Smoke Amount	Tobacco Type
pH: non-stress	0.00 (0.06, *N* = 41)	−0.00 (0.05, *N* = 41)	−0.00 (0.08, *N* = 41)
TAC: non-stress	−0.00 (0.09, *N* = 41)	0.01 (0.03, *N* = 41)	0.01 (0.04, *N* = 41)
STAI: nonstress	−0.00 (0.09, *N* = 41)	0.03 * (0.00, *N* = 41)	0.03 (0.05, *N* = 41)
pH: stress	−0.09 (0.05, *N* = 41)	−0.09 (0.05, *N* = 41)	0.02 (0.09, *N* = 41)
TAC: stress	0.01 (0.04, *N* = 41)	0.03 (0.08, *N* = 41)	−0.00 (0.09, *N* = 41)
STAI: stress	−0.01 (0.04, *N* = 41)	−0.07 (0.06, *N* = 41)	0.01 (0.03, *N* = 41)

Pearson’s correlation, sig. (2-tailed), *N*; * *p* < 0.05.

## Data Availability

The data presented in this study are available upon request from the corresponding author. The data are not publicly available in accordance with the consent provided by participants on the use of confidential data.

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
