# Peer review of "Total Antioxidant Capacity of Saliva and Its Correlation with pH Levels among Dental Students under Different Stressful Conditions"

_diagnostics, 2023, doi:10.3390/diagnostics13243648_

Round 1
Reviewer 1 Report (Previous Reviewer 1)
Comments and Suggestions for Authors
Authors have satisfactorily addressed all comments
Author Response
Dear Reviewer,
We appreciate your thorough review of our article and are pleased to hear that you find the authors have satisfactorily addressed all comments. We are committed to ensuring that our work meets the highest standards, and your input is highly valued.
Thank you for your time and consideration.
Reviewer 2 Report (Previous Reviewer 2)
Comments and Suggestions for Authors
1. pH was measured using test strips, so the accuracy of pH determination is 6, 6.5, 7. The authors calculate the average value accurate to two decimal places and conclude that during the non-stressful period, the average pH value recorded was 6.73, while during the stressful period, the average pH value was 6.58. On what basis? The authors cannot give a number more accurate than the original data, i.e. it is necessary to round to whole numbers and the differences will be insignificant.
2. The list of references still contains many old sources; the authors have not revised the list.
3. In general, I would like to see responses to the reviewer's comments in the previous round of review.
4. Why are tables needed? Information can be given directly in the text. The drawings are of low quality, the inscriptions are not readable. In general, if we are talking about comparing groups, we would like to see graphs and comparisons of parameters with whiskers and p-values.
Author Response
- pH was measured using test strips, so the accuracy of pH determination is 6, 6.5, 7. The authors calculate the average value accurate to two decimal places and conclude that during the non-stressful period, the average pH value recorded was 6.73, while during the stressful period, the average pH value was 6.58. On what basis? The authors cannot give a number more accurate than the original data, i.e. it is necessary to round to whole numbers and the differences will be insignificant.
Response
Dear Reviewer,
Thank you for your thorough review of our article. We appreciate your keen observation regarding the pH values reported in the draft.
During the drafting process, the mean and median values have been calculated and shown in Table 3. Maybe, this was not clear in the text description and this unintentional error resulted in a misleading reporting of values accurate to two decimal places instead of one. We sincerely apologize for any confusion this may have caused. We changed the phrasing in the paragraph to make it clearer.
- The list of references still contains many old sources; the authors have not revised the list.
Response
Dear Reviewer,
We appreciate your diligence in reviewing the references, and we have carefully considered your suggestion.
In response to your comment, we would like to clarify that in the initial revision, we removed two of the oldest references as per your recommendation. However, upon further reflection and taking into account your guidance, we have extensively revised the entire bibliography. All sources older than 10 years have been eliminated to ensure the most current and relevant references are included. We believe this adjustment enhances the overall quality of our article and aligns with your suggestion to update the reference list.
- In general, I would like to see responses to the reviewer's comments in the previous round of review.
Response
The article made a mixed impression.
- Methodological issues: pH was measured using test strips - this is not serious, since the difference in pH between groups may be in the second decimal place, which cannot be recorded with a test strip.
RESPONSE
Dear Reviewer,
We appreciate your feedback regarding the methodological aspect of our study.
The concern you raised about pH measurement using test strips is duly noted. We would like to clarify that while pH measurement using test strips may not provide high precision, it has been employed in various studies and real-world applications. While higher precision instruments are available, practical considerations sometimes lead researchers to opt for pH test strips. These test strips are readily accessible and can offer a reasonable estimation of pH levels, especially in cases where highly precise measurements are not required. Furthermore, it's important to consider the practical aspects of our study. The convenience and simplicity of using test strips were advantageous in our case, given the nature of our research and the resources available. We acknowledge the limitations of this method in terms of precision, and we have discussed these limitations in our manuscript. While we understand the concern, we believe that the method provided a suitable compromise between practicality and precision for our specific research objectives.
- Table in line 334 without title and number: a complete mess with significant figures, missing points in the standard deviation. The graphs show that the distribution is not normal, why is the standard deviation calculated? Has the nature of the distribution been checked?
RESPONSE
Dear Reviewer,
The distribution has been checked by calculating the value of kurtosis for each scale we used in both condition (stress present/non-stress). We considered a normal distribution for the scale because as stated in the book Laboratory Statistics Methods in Chemistry and Health Sciences, Second Edition, 2018, a standard normal distribution has kurtosis of 3 and is recognized as mesokurtic. This was the rationale behind calculating the standard deviation as well.
- In general, I do not see the novelty and relevance of the work at the level of a school research project. Please note that almost all of the publications in the bibliography are earlier than 2015, there are almost no recent ones, which indicates poor elaboration of the issue.
RESPONSE
Dear Reviewer,
Considering that this is fundamentally an exploratory study, we opted for the most feasible sample. Regarding the literature review, we selected what we deemed the most relevant publications. In response to the received review, we have revised and removed the outdated bibliography. However, we remain open to updating and revising the data and bibliography based on the feedback received.
- Why are tables needed? Information can be given directly in the text. The drawings are of low quality, the inscriptions are not readable. In general, if we are talking about comparing groups, we would like to see graphs and comparisons of parameters with whiskers and p-values.
Response
Dear Reviewer,
Tables are crucial in presenting complex data succinctly, allowing readers to easily compare and analyze information. While information can be included in the text, tables provide a structured format for presenting detailed data, making it more accessible and facilitating a comprehensive understanding.
We acknowledge your concerns regarding the low quality of drawings and illegible inscriptions. In response, we have replaced the figures with higher quality visuals that aim to improve clarity and readability. We believe these revisions will significantly enhance the overall visual presentation of our findings.
Round 2
Reviewer 2 Report (Previous Reviewer 2)
Comments and Suggestions for Authors
It seems that the authors do not hear my comments. The methodological part states: “Salivary pH values underwent changes during the stressful period. Therefore, the majority of participants had a pH of 6.5 (31 subjects), while 25 students had a pH of 7, three subjects had a pH of 5.5, 16 of they had a pH equal to 6, and 5 subjects had a pH of 7.5." Thus, pH has discrete values of 5.5, 6.0, 6.5, 7.0, 7.5. When the authors calculate group averages, they obtain values that are an order of magnitude more accurate: 6.73 and 6.58. I asked where the extra decimal place came from, but received no answer. According to the rules, we cannot obtain a value more accurate than the original data during calculations, which means that after calculations the authors must round the values to 6.7 and 6.6 at least. Or use not the average values, but the medians, which are given in Table 3.
Regarding the tables, they duplicate the information given in the text of the manuscript. Table 1 - lines 197-199, Table 2 - lines 289-291. You need to leave one thing, and I recommend removing the tables, they are not informative.
Author Response
Dear reviewer, thank you for your prompt response and the careful analysis of our manuscript. We are sorry that you felt, we were not able to clarify all the issues that came up. For this reason, we are going to try to explain our statistical analysis in this paragraph.
In mathematics and statistics, the arithmetic mean, arithmetic average, or just the mean or average is the sum of a collection of numbers divided by the count of numbers in the collection. Therefor the mean PH value was calculated according to the formula in the attached word document.
2. Regarding the tables, they duplicate the information given in the text of the manuscript. Table 1 - lines 197-199, Table 2 - lines 289-291. You need to leave one thing, and I recommend removing the tables, they are not informative.
Response: Although we feel that tables are crucial in presenting complex data succinctly, allowing readers to easily compare and analyze information, we removed tables 1 and 2 as you suggested.
We hope this answers suit your demands

Round 3
Reviewer 2 Report (Previous Reviewer 2)
Comments and Suggestions for Authors
The authors again do not understand the remark that I have already made several times. The accuracy of pH measurement using test strips allows you to determine the pH approximately (5.5, 6.0, etc.). No calculations of the mean and median can give an accuracy comparable to a laboratory pH meter (6.78 and 6.53 for example), so it is necessary to correct the values in Table 1 and leave only one decimal place.
According to the text, if you write that 1 person had a pH of 8, then this means that the accuracy of the determination is even lower; it must be corrected to 8.0 everywhere. Only in this case do we even have the right to talk about the first decimal place. If you write 6, 7, 8 (instead of 6.0, 7.0, 8.0), then all results should be rounded to whole numbers, but then there is no difference between groups.
The authors incorrectly process experimental data and mislead the reader. It is not possible to assess subtle differences between subgroups with this method. This is a gross methodological violation.
There are no units specified for TAC.
Author Response
The authors again do not understand the remark that I have already made several times. The accuracy of pH measurement using test strips allows you to determine the pH approximately (5.5, 6.0, etc.). No calculations of the mean and median can give an accuracy comparable to a laboratory pH meter (6.78 and 6.53 for example), so it is necessary to correct the values in Table 1 and leave only one decimal place.
R: Referring to the table 3 the meaning for showing also mean and median values was to show full transparency for the data collected. Because it wasn't feasible at the time to use another type of measurement tool, fully aware of the fact that test strips only give approximate values of the pH level, we used indicators of correlation and measured the significance value as a more explicit approach.
According to the text, if you write that 1 person had a pH of 8, then this means that the accuracy of the determination is even lower; it must be corrected to 8.0 everywhere. Only in this case do we even have the right to talk about the first decimal place. If you write 6, 7, 8 (instead of 6.0, 7.0, 8.0), then all results should be rounded to whole numbers, but then there is no difference between groups.
There are no units specified for TAC.
R: Thank you for pointing out the accuracy of the reported values. The authors corrected the pH values and added the measurement units for TAC where appropriate in the text.
Round 4
Reviewer 2 Report (Previous Reviewer 2)
Comments and Suggestions for Authors
The authors made changes to the manuscript in accordance with the reviewer's comments. I believe that in its present form the manuscript can be recommended for publication.
This manuscript is a resubmission of an earlier submission. The following is a list of the peer review reports and author responses from that submission.
Round 1
Reviewer 1 Report
Comments and Suggestions for Authors
Comments for authors
1. Methodology - Authors should mention the inclusion and exclusion criteria for the sampling
2. Methodology - Do students with history of mental health also included in the study?
3. Authors should provide sample size calculation for the study
4. Methodology - Were the questionnaire used are in English or translated?
Reviewer 2 Report
Comments and Suggestions for Authors
The article made a mixed impression.
1. Methodological issues: pH was measured using test strips - this is not serious, since the difference in pH between groups may be in the second decimal place, which cannot be recorded with a test strip.
2. Table in line 334 without title and number: a complete mess with significant figures, missing points in the standard deviation. The graphs show that the distribution is not normal, why is the standard deviation calculated? Has the nature of the distribution been checked?
3. In general, I do not see the novelty and relevance of the work at the level of a school research project. Please note that almost all of the publications in the bibliography are earlier than 2015, there are almost no recent ones, which indicates poor elaboration of the issue.